# Real-Time Heart Rate Detection Method Based on 77 GHz FMCW Radar

**DOI:** 10.3390/mi13111960

**Published:** 2022-11-11

**Authors:** Xiaohong Huang, Zedong Ju, Rundong Zhang

**Affiliations:** 1College of Artificial Intelligence, North China University of Science and Technology, Tangshan 063210, China; 2Hebei Key Laboratory of Industrial Intelligent Perception, Tangshan 063210, China; 3College of Management, North China University of Science and Technology, Tangshan 063210, China

**Keywords:** real-time heart rate, 77 GHz FMCW radar, random body motion, multi-detection-point adaptive harmonics cancellation, linear predictive coding

## Abstract

This paper proposes a real-time heart rate detection method based on 77 GHz FMCW radar. Firstly, the method establishes a new motion model according to respiratory and heartbeat rules, and extracts the motion signals of the chest and the abdomen; then, the random body motion (RBM) signal is eliminated by a combination of polynomial fitting and recursive least squares (RLS) adaptive filtering; lastly, multi-detection-point adaptive harmonics cancellation (AHC) is used to eliminate respiratory harmonics. In addition, the method introduces a spectrum analysis algorithm based on linear predictive coding (LPC). The experimental results show that the method can effectively eliminate the RBM signal and respiratory harmonics, and that the average real-time heart rate detection error rate is 2.925%.

## 1. Introduction

Heart rate is one of the important vital signs parameters: it can not only reflect the health status of the human body, but can also provide a reference for clinical diagnosis [1,2]. Furthermore, heart rate variability can reflect the activity, balance and related pathological states of the cardiac autonomic nervous system [3]. As a non-contact detection instrument, Doppler radar can detect heartbeat signals from micro-motion signals on the surface of the human body, and it has attracted more and more attention in the medical field [4,5,6,7].

However, there are two problems for Doppler radar, in detecting heart rate in near real-time: the first is how to eliminate the RBM signal, which seriously affects the accuracy of heart rate extraction [8]; the second is how to eliminate respiratory harmonics. Respiratory harmonics and heartbeat are relatively close in frequency; therefore, respiratory harmonics may be mistaken for heartbeat signals [9].

In references [10,11,12], the RBM signal was eliminated using two systems; however, it was difficult for the two systems to operate synchronously without interfering with one other. Gu et al. [13] proposed a method to remove the RBM signal using deep neural networks, which successfully extracted the respiration rate; however, it was difficult to extract the heart rate due to lack of training data. Yang et al. [14] proposed a scheme combining adaptive noise cancellation with polynomial fitting, which would effectively eliminate the RBM signal; however, the step size factor was fixed in the adaptive noise cancellation algorithm, which made it difficult to balance steady-state error and convergence speed.

In references [15,16], to eliminate respiratory harmonics, the respiratory harmonics cancellation method and the adaptive harmonics comb notch digital filter method were proposed; however, both these methods required the respiratory rate to eliminate the harmonics, which increased the dependence of the heart rate on the respiratory rate. In reference [17], to distinguish between respiratory harmonics and heartbeat signals, the heartbeat signals were amplified; however, the respiratory harmonics were also amplified, which reduced the accuracy of the heart rate extraction. Yang et al. [18] proposed a wave-form-driven matched filtering method based on polynomial fitting; however, it was difficult to extract the heart rate quickly.

In addition to the above problems, real-time heart rate detection also needs to solve the problem of insufficient frequency resolution. To this end, Lee et al. [19] proposed an algorithm based on redistribution of the joint time–frequency transform. Hu et al. [20] applied a continuous wavelet filter and ensemble empirical mode decomposition to recover and separate cardiopulmonary signals; they used the peak-to-peak interval to detect the frequency. References [5,21] proposed the time-window-variation technique combined with the fast Fourier transform and wavelet transform, respectively. Park et al. [22] proposed a novel polyphase-basis discrete cosine transform. Ye et al. [23] proposed a stochastic gradient algorithm based on the time-window-variation technique. Lv et al. [24] proposed a non-contact short-term heart rate detection system based on a 120 GHz narrow beam FMCW millimeter wave radar. Gao et al. [25] proposed a method for extracting the life activity spectrum based on millimeter wave radar; although this method was able to extract the heart rate in near real-time, the acquisition time was long, and the accuracy of the heart rate left room for improvement.

In order to solve the above problems, this paper proposes a real-time heart rate detection method based on 77 GHz FMCW radar, and conducts experimental verification. The method establishes a new motion model according to respiratory and heartbeat rules, eliminates the RBM signal by a combination of polynomial fitting and RLS adaptive filtering, and eliminates respiratory harmonics by multi-detection-point AHC. In addition, a spectrum analysis algorithm based on LPC has been introduced, to improve detection accuracy.

## 2. Theory

Figure 1 shows the block diagram of a typical FMCW radar structure. The transmitted signal can be expressed as:(1)$$S(t)=\cos(2\pi f_{c}t+\pi \frac{B}{T_{c}}t^{2})$$
where $f_{c}$ is the carrier frequency, B is the total bandwidth and $T_{c}$ is the duration. Assume the initial distance between the radar and the body is $d_{0}$; then, the received signal can be expressed as:(2)$$R(t)=\cos\left\{2\pi f_{c}\left[t-\frac{2\left[d_{0}+x(t)\right]}{c}\right]+\pi \frac{B}{T_{c}}{\left[t-\frac{2\left[d_{0}+x(t)\right]}{c}\right]}^{2}\right\}$$
where c is the speed of light, and x(t) is the motion of the human body; then, the intermediate frequency signal can be expressed as:(3)$$R_{IF}(t)=\exp\left\{j\left[2\pi \frac{2Bd_{0}}{cT_{c}}t+\frac{4\pi \left[d_{0}+x(t)\right]}{\lambda }\right]\right\}$$
where λ is the maximum wavelength of the signal. The sampled signal can be expressed as:(4)$$R_{IF}(mT_{s}+nT_{f})=\exp\left\{j\left[2\pi \frac{2Bd_{0}}{cT_{C}}nT_{f}+\frac{4\pi \left[d_{0}+x(mT_{s})\right]}{\lambda }\right]\right\}$$
where m and n represent the index number of chirps and the index number of the sampling point in each chirp, respectively; $T_{s}$ is the slow time sampling period, and $T_{f}$ is the fast time sampling period.

## 3. Methods

### 3.1. New Motion Model

Figure 2 shows the structure of the chest and diaphragm. When you inhale, the chest and diaphragm expand outward. When you exhale, the chest and diaphragm contract inward. The motion of the chest can be expressed as:(5)$$x_{1}(t)=a_{r}\sin(2\pi f_{r}t)+a_{h}\sin(2\pi f_{h}t)+n_{t}(t)$$
where $a_{r}$ is the thoracic amplitude due to respiration, $f_{r}$ is the respiration rate, $a_{h}$ is the thoracic amplitude due to heartbeat, $f_{h}$ is the heart rate, and $n_{t}(t)$ is the noise signal; $n_{t}(t)$ contains the RBM signal. The motion of the abdomen can be expressed as:(6)$$x_{2}(t)=a_{a}\sin(2\pi f_{r}t)+n_{a}(t)$$
where $a_{a}$ is the abdominal amplitude due to respiration, and $n_{a}(t)$ is the noise signal; $n_{a}(t)$ contains the RBM signal.

Assume that the initial distance from the radar to the chest and abdomen are $d_{1}$ and $d_{2}$, respectively. The chest and abdomen belong to the far field targets. After introducing the multi-receiving antenna model, Formula (4) can be rewritten as:(7)$$R_{IF}(mT_{s}+nT_{f},l)=\sum_{k=1}^{2}\exp\left\{j\left[2\pi \frac{2Bd_{k}}{cT_{C}}nT_{f}+\frac{4\pi \left[d_{k}+x_{k}(mT_{s})\right]}{\lambda }+\frac{2\pi (l-1)d\sin\theta_{k}}{\lambda }\right]\right\}$$
where l represents the index number of receiving antennas, d is the distance between adjacent receiving antennas, and $\theta_{1}$ and $\theta_{2}$ are the respective azimuth angles from the chest and from the abdomen to the radar.

### 3.2. Acquisition of Vital Signs

This paper, to obtain the distance-Doppler-angle dimension data, performed fast time dimension FFT, slow time dimension FFT and angle calculation on $R_{IF}(mT_{s}+nT_{f},l)$. The Doppler dimensional effective spectral lines were calculated according to the rules of respiration and heartbeat. The formula for calculating the lower and upper spectral lines can be expressed as:(8)$$p_{1}=\left\{\min\left(\frac{4MT_{a}a_{r}}{t_{r-\max}\lambda },\frac{4MT_{a}a_{a}}{t_{r-\max}\lambda }\right)\right\}$$
(9)$$p_{2}=\left\{\max\left(\frac{4MT_{a}a_{r}}{t_{r-\min}\lambda },\frac{4MT_{a}a_{a}}{t_{r-\min}\lambda }\right)\right\}$$
where M is the number of chirps, $T_{a}$ is the time interval, $t_{r-\max}$ is the maximum respiratory cycle, and $t_{r-\min}$ is the minimum respiratory cycle. The energy of the Doppler slices was accumulated between the Doppler dimensions [$p_{1}$–$p_{2}$]; the energy value of each point can be expressed as:(10)$$E\left(r,a\right)=\sum_{p=p_{1}}^{p_{2}}M_{r,p,a}\left(r,p,a\right)$$
where $M_{r,p,a}$ is distance, Doppler slice, and angle dimension data. Finally, the data after energy accumulation was used for distance and angle dimension CFAR detection.

### 3.3. Elimination of the RBM Signal

Figure 3 shows the block diagram of the RBM signal elimination. A polynomial fitting algorithm was used to fit the RBM signal from $x_{i}(n)$, denoted ${\hat{x}}_{i}(n)$. Assuming that the tap-weight vector of the digital filter is $w_{i}(n-1)$, then the estimation error vector can be expressed as:(11)$$e_{i}(n)=x_{i}(n)-{\hat{y}}_{i}(n)=x_{i}(n)-w_{i}{\left(n-1\right)}^{T}{\hat{x}}_{i}(n)$$
where T denotes transposition. The error-weighted sum of the squares of $e_{i}(n)$ can be expressed as:(12)$$s_{i}\left(n\right)=\sum_{k=1}^{n}\sigma^{n-k}{\left|e_{i}(n)\right|}^{2}$$
where σ is the forgetting factor. At this point, the tap-weight vector can be expressed as:(13)$$w_{i}(n)=w_{i}(n-1)+R_{i}^{-1}(n){\hat{x}}_{i}(n)e_{i}(n)$$
where $R_{i}^{-1}(n)$ is a non-fixed step factor. When the value of $s_{i}\left(n\right)$ is minimum, the cycle will end and the estimation error vector $e_{i}(n)$ is output.

### 3.4. Elimination of Respiratory Harmonics

Figure 4 shows the block diagram of respiratory harmonics elimination. The motion signal of the chest after removing the RBM signal is denoted by $e_{1}(n)$, which contains respiratory signals, respiratory harmonics and heartbeat signals. The motion signal of the abdomen after removing the RBM signal is denoted $e_{2}(n)$, which contains respiratory signal and respiratory harmonics. The second, third and fourth harmonics of respiration are easily confused with the heartbeat signal; therefore, $e_{1}(n)$ and $e_{2}(n)$ are decomposed into five IMF (i.e., $IMF_{1}1$, $IMF_{1}2$, $IMF_{1}3$, $IMF_{1}4$ and $IMF_{1}5$) and four IMF (i.e., $IMF_{2}1$, $IMF_{2}2$, $IMF_{2}3$ and $IMF_{2}4$), respectively. Signal components with frequencies between [0.6–2.5 Hz] are reconstructed separately and denoted $h_{t,r}^{\prime }(n)+h_{t,h}(n)$ and $h_{a,r}^{\prime }(n)$. When performing harmonic elimination processing, $h_{t,r}^{\prime }(n)+h_{t,h}(n)$ and $h_{a,r}^{\prime }(n)\cdot a_{c}$ are used as the original signal and the reference signal, respectively (the adaptive filtering process based on RLS is detailed in Section 3.3); $a_{c}$ is the energy weight, which can be expressed as:(14)$$a_{c}=\max\left\{E\left[h_{t,r}^{\prime }(n)+h_{t,h}(n)\right]\right\}/\max\left\{E\left[h_{a,r}^{\prime }(n)\right]\right\}$$
where $E\left[\;\right]$ represents the energy of each signal component; when the cycle ends, an approximate heartbeat signal can be obtained. Using LPC [26] to extend $h_{t,h}^{\prime }(n)$, it can be expressed as:(15)$${\hat{h}}_{t,h}^{\prime }(n)=\sum_{l=1}^{L}a_{l}h_{t,h}^{\prime }(n-l)$$
where L is the model order, and $a_{l}$ is the prediction coefficient. Finally, analyzing the spectrum of ${\hat{h}}_{t,h}^{\prime }(n)$ can obtain the heart rate.

## 4. Results and Discussion

We conducted the experimental validation using the commercial Texas Instruments IWR1443BOOST mmWave radar sensor (Texas Instruments, Dallas, TX, USA), which can form an FMCW radar in a MIMO system. The system parameters were:$f_{c}=77\;\mathrm{GHz}$; B=3.98 GHz; $T_{c}=52\;\mu s$; λ=3.9 mm; N=512; $T_{f}=0.07\;\mu s$; M=256; $T_{s}=50\;ms$. Figure 5 shows the experimental scene. During data collection, each subject sat directly in front of the radar, typing on the keyboard with his/her left hand while moving the mouse with his/her right hand. The subject’s left and right arms performed irregular movements, and the subject’s chest and abdomen slowly shook back and forth, which simulated the motion state of office workers in a real environment as far as possible. At the same time, there was a medical finger-pressure pulse sensor YX306 (Yuwell, Suzhou, China) on the finger, which displayed the number of heartbeats. When the value of the figure-pressure pulse sensor was stable, the radar started to collect data and record the value displayed by the finger-pressure pulse sensor every second (reference heart rate). As the data collected by the radar and the numerical value recorded by the figure-pressure pulse sensor needed to be synchronized, this may also have caused small errors. Additionally, we measured the distance with a tape measure, and calculated the angle (expected result). In order to verify the real-time heart rate extraction performance, we collected data for 12.8 s each time, and implemented the method in MATLAB. To quantify the performance of the method, the error rate and the average error rate were introduced into the performance analysis, which can be expressed as:(16)$$error=\left|\frac{HR_{mea}-HR_{rea}}{HR_{rea}}\right|\times 100\%$$
(17)$$\bar{\mathrm{error}}=\left(\sum_{l=1}^{L}error_{l}\right)/L$$
where $HR_{mea}$ is the measured heart rate, $HR_{rea}$ is the average heart rate recorded by the figure-pressure pulse sensor, and L is the number of measurement groups. It should be noted that the radar board was placed vertically, and clockwise was positive.

Figure 6 shows a set of experimental results. The chest and abdomen belong to the far field targets. Figure 6a shows the localization results of the chest and abdomen, the distance and angle of which were: $21^{{}^{\circ}}$, 45.91 cm, $44^{{}^{\circ}}$ and 28.69 cm. These results were in fair agreement with the expected values. Figure 6b,c show the motion signals of the chest and abdomen, respectively, after removing the RBM signal. As can be seen, the RBM signal has been removed and the baseline is effectively back on the central axis. To better demonstrate the performance of real-time heart rate extraction, the 12.8 s signal was equally divided into four segments (i.e., segment 1, segment 2, segment 3, segment 4). Figure 6d–g show the results of the four-segment elimination of the respiratory harmonics. As can be seen from the figure, the motion signals of the chest and abdomen contained respiratory harmonics of the same frequency. Observing the results, it can be seen that the respiratory harmonics were effectively eliminated. The signal after harmonic elimination was linearly predicted based on LPC, and the heart rates obtained by spectrum analysis were: 1.527 Hz, 1.464 Hz, 1.412 Hz and 1.465 Hz. The error rates were: 3.74%, 1.74%, 0.07% and 3.83%, respectively. The average error rate was 2.35% and the computer calculation time was 3.01 s (MagicBook 16 Pro).

We validated the proposed approach by conducting 15 experiments on five subjects, differing in height (172–188 cm) and in age (26–49 years). During the experiment, each subject sat directly in front of the radar and used his/her computer. We collected data for 12.8 s each time. Table 1 shows the results of the experimental validation. The average error rates of the four segments were: 3.10%, 3.37%, 2.68% and 2.55%. The average error rate for all experiments was 2.925%. At the same time, the computer running time was less than 3.01 s. This shows that the proposed method can effectively extract the real-time heart rate.

Table 2 lists the comparison results with other works. We set the distance between 0.28–0.70 m to simulate an office scene realistically. When detecting the real-time heart rate, the time window length was generally required to be less than 5 s. In order to better verify the performance of the proposed method in detecting the real-time heart rate, we set a 3.2 s time window for detecting the heart rate. In general, the shorter the time window, the lower the accuracy of the heart rate detection. As can be seen from the table, the biggest advantage of the proposed method is that the required detection time is short, and accurate heart rate detection can be performed at any time.

## 5. Conclusions

In order to improve the application of non-contact detection technology in the medical field, this paper proposes a real-time heart rate detection method based on 77 GHz FMCW radar. The method eliminates the RBM signal by a combination of polynomial fitting and RLS adaptive filtering, and eliminates respiratory harmonics by a multi-detection-point AHC. The heart rate is then obtained using an LPC-based spectral analysis algorithm. Through experiments, it was found that the proposed method effectively eliminated the RBM signal and respiratory harmonics, and detected the real-time heart rate of office workers. The proposed method has great potential for the real-time heart rate detection of key personnel, such as office workers, drivers, etc.

## Figures and Tables

**Figure 1 micromachines-13-01960-f001:**
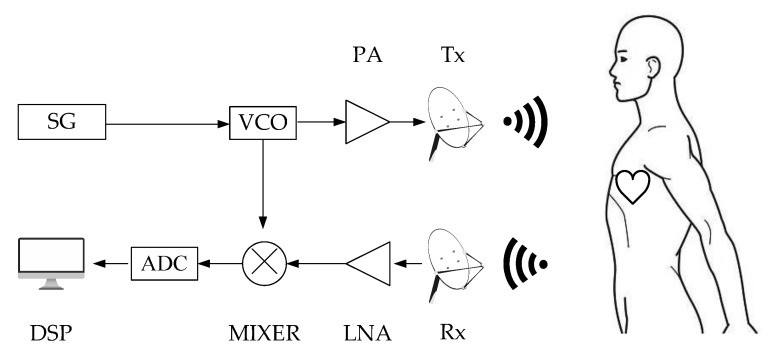
Block diagram of the non-contact FMCW radar. SG: Signal Generator; VCO: Voltage Controlled Oscillator; PA: Power Amplifier; Tx: Transmitting antenna; Rx: Receiving antenna; LNA: Low Noise Amplifier; MIXER: Mixer; ADC: Analog Digital Converter; DSP: Digital Signal Processing.

**Figure 2 micromachines-13-01960-f002:**
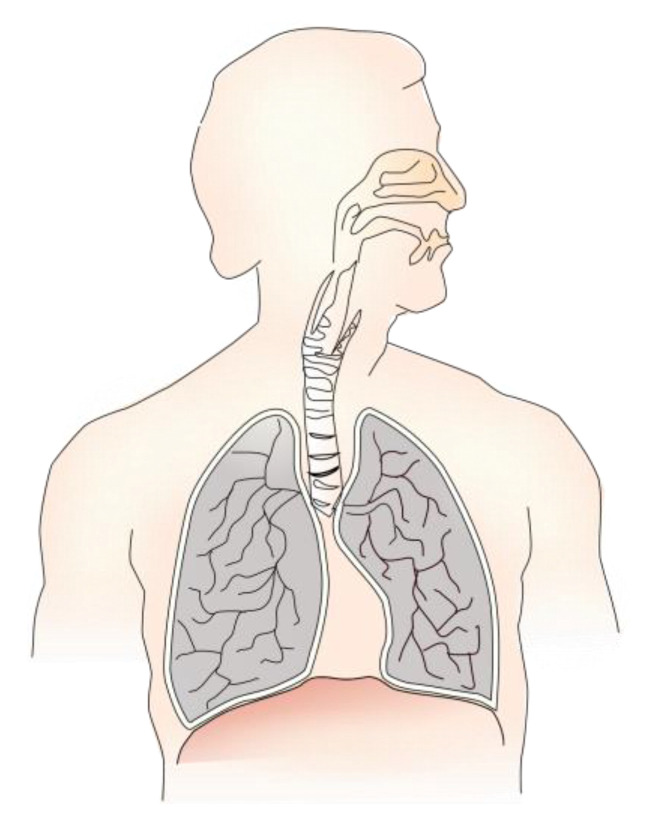
Structure of the chest and diaphragm.

**Figure 3 micromachines-13-01960-f003:**
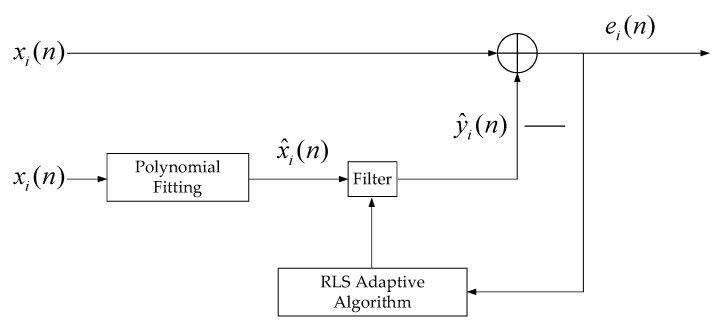
Block diagram of the RBM signal elimination: $x_{i}(n)$ is the motion signal of the human body; RLS: recursive least squares; $x_{1}(n)$ is the motion signal of the chest; $x_{2}(n)$ is the motion signal of the abdomen.

**Figure 4 micromachines-13-01960-f004:**
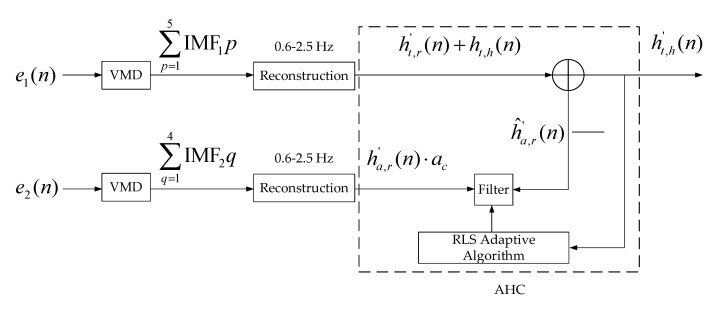
Block diagram of respiratory harmonics signal elimination: $e_{1}(n)$ is the motion signal of the chest after removing the RBM signal; $e_{2}(n)$ is the motion signal of the abdomen after removing the RBM signal. VMD: Variational Mode Decomposition; IMF: Intrinsic mode function; AHC: adaptive harmonics cancellation.

**Figure 5 micromachines-13-01960-f005:**
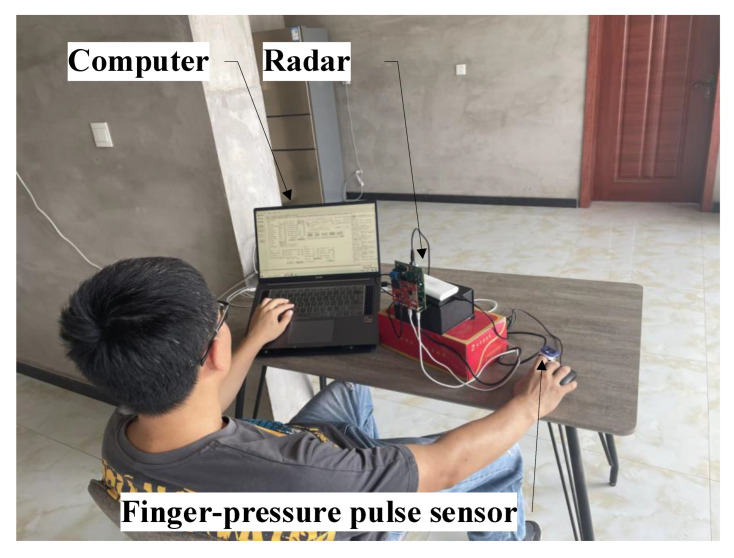
Experimental environment.

**Figure 6 micromachines-13-01960-f006:**
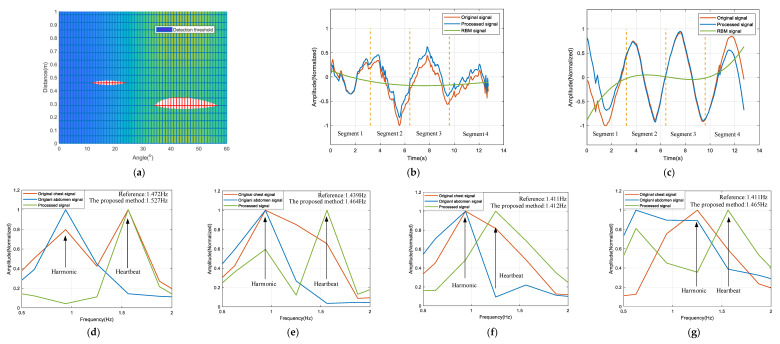
Experimental results: (**a**) localization results of the chest and abdomen; (**b**) the motion signal of the chest after removing the RBM signal; (**c**) the motion signal of the abdomen after removing the RBM signal; (**d**) the result of the segment 1 elimination of respiratory harmonics; (**e**) the result of the segment 2 elimination of respiratory harmonics; (**f**) the result of the segment 3 elimination of respiratory harmonics; (**g**) the result of the segment 4 elimination of respiratory harmonics.

**Table 1 micromachines-13-01960-t001:** Results of the experimental validation.

	Segment 1 (3.2 s)			Segment 2 (3.2 s)			Segment 3 (3.2 s)			Segment 4 (3.2 s)		
	$HR_{mea}/\mathrm{Hz}$	$HR_{rea}/\mathrm{Hz}$	error/%	$HR_{mea}/\mathrm{Hz}$	$HR_{rea}/\mathrm{Hz}$	error/%	$HR_{mea}/\mathrm{Hz}$	$HR_{rea}/\mathrm{Hz}$	error/%	$HR_{mea}/\mathrm{Hz}$	$HR_{rea}/\mathrm{Hz}$	error/%
1	1.5124	1.5722	3.80	1.6625	1.5833	5.00	1.4808	1.5222	2.72	1.4584	1.4583	0.01
2	1.4452	1.4778	2.21	1.4461	1.4500	0.27	1.4387	1.4167	1.56	1.4293	1.4167	0.89
3	1.6998	1.6333	4.07	1.6437	1.5944	3.09	1.5625	1.5222	2.65	1.4323	1.4250	0.51
4	1.4176	1.3500	5.01	1.3715	1.3222	3.29	1.4168	1.3278	6.70	1.3697	1.3611	0.61
5	1.5276	1.4667	4.15	1.4638	1.4389	1.73	1.4123	1.4111	0.08	1.4650	1.4056	3.82
6	1.3385	1.3222	1.23	1.4446	1.3222	9.36	1.4305	1.3167	8.65	1.3753	1.3500	1.87
7	1.4252	1.4417	1.14	1.4235	1.4167	0.48	1.3939	1.4000	0.44	1.4058	1.3722	2.45
8	1.4366	1.3833	3.85	1.3347	1.3833	3.52	1.4048	1.3944	0.74	1.3410	1.3833	3.06
9	1.4153	1.4111	0.29	1.4159	1.4500	2.35	1.4047	1.3792	1.85	1.3329	1.3125	1.55
10	1.6660	1.5778	5.59	1.6342	1.5667	4.31	1.4657	1.4917	1.74	1.3995	1.4444	3.11
11	1.3705	1.4500	5.48	1.4312	1.4111	1.42	1.3589	1.3667	0.57	1.3481	1.3722	1.76
12	1.4006	1.4056	0.35	1.4104	1.4444	2.80	1.3733	1.3889	1.28	1.4198	1.3400	5.96
13	1.4215	1.4667	3.08	1.4167	1.4500	2.30	1.3647	1.3889	1.74	1.3864	1.3667	1.44
14	1.3649	1.3000	4.99	1.3854	1.3000	6.57	1.3966	1.3056	6.97	1.3611	1.2944	5.15
15	1.4124	1.3944	1.29	1.3930	1.3389	4.04	1.3668	1.3333	2.51	1.3897	1.3222	5.10
	$\bar{\mathrm{error}}$/%		3.10	$\bar{\mathrm{error}}$/%		3.37	$\bar{\mathrm{error}}$/%		2.68	$\bar{\mathrm{error}}$/%		2.55

**Table 2 micromachines-13-01960-t002:** Comparison of the proposed method with the other existing real-time heart rate detection methods.

Ref. No.	Detection Distance/m	Acquisition Time/s	Time Window Length/s	$\bar{\mathrm{error}}/\%$
[19]	0.5	75	5	5
[20]	0.5	240	15	<3
[21]	Not mentioned	30	2–5	3.4
[5]	Not mentioned	60	3–5	3.5
[22]	1–1.1	80–90	3, 2, 1.5	3.75, 5.58, 7.58
[23]	0.8	120	8	4.2
[24]	1	100	3	4.38
[25]	<0.5	Not mentioned	Not mentioned	3.65
This work	0.28–0.70	12.8	3.2	2.925

## Data Availability

The data that support the plots within this paper, and other findings of this study, are available from the second author upon reasonable request.

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
