# Peer review of "Real-Time Heart Rate Detection Method Based on 77 GHz FMCW Radar"

_micromachines, 2022, doi:10.3390/mi13111960_

Round 1
Reviewer 1 Report
This paper addresses noncontact heart-rate detection with a state-of-the-art 77 GHz FMCW radar. The paper is well written with theory and methods of random body motion suppression and respiratory harmonics suppression which can interfere with heart rate detection; provides experimental results with a commercial miniature MIMO radar; and shows improvement over past work in accuracy and time of detection.
Major suggestions:
1. The paper may include the effect of prescribed motion of the subject sitting on a chair by moving the chair sideways and front/back. This is important from the standpoint of nonstationary targets.
2. Additionally, ascertain whether the subject is in the far field of the antennas to avoid near field effects.
Below are some minor suggestions:
1. Denote IF signal with a different symbol from B(t) as B is used for bandwidth
2. In Eq. (7), i should be changed to l
3. Line 113, date should be changed to data
4. It is better to change Parts 1-4 etc. to segments 1-4 in Fig. 6
Reviewer 2 Report
Algorithm for estimating heart rate using FMCW rader is well designed. In the verification of the results, it seems that the value measured by the FMCW radar sensor and the value of the finger pressure pulse sensor were compared. HRmea and HRrea need to be expressed concretely and clearly.
It is also necessary to prove the finger perssure pulse sensor.
Reviewer 3 Report
The paper sounds overall timely and of possible interest for the readers of micromachines. The manuscript proposes a real-time heart rate detection method. The method eliminates the RBM signal by a combination of polynomial fitting and RLS adaptive filter, and eliminates respiratory harmonics by a multi-detection-point AHC. Manuscript organized each sections clearly to convey the scope of research. The formulas and charts in the manuscript are clearly described, which supports the author's work well and and has an overall acceptable shape.
1) Please try to explain the setting basis of the division of detection distance and time window length in Table 2 and the impact on the detection results.
2) There are few references in this paper in recent three years, which can not fully reflect the current research status. It is suggested to supplement relevant references.
3) There are two kinds of problems in real-time heart rate detection by Doppler radar mentioned in the manuscript, including how to eliminate the RBM signal and eliminate respiratory harmonics. Whether there are similar problems in the FMCW radar used in the author's work, it is suggested to supplement.
Round 2
Reviewer 2 Report
The request has been well modified.